# Revisiting 3D Object Detection From an Egocentric Perspective

**Boyang Deng**[†] [*]  **Charles R. Qi**[†]  **Mahyar Najibi**[†]

**Thomas Funkhouser**[‡]  **Yin Zhou**[†]  **Dragomir Anguelov**[†]

[†]**Waymo LLC**  [‡]**Google Research**

## Abstract

3D object detection is a key module in safety-critical robotics applications such as autonomous driving. For such applications, we care the most about how the detections impact the ego-agent's behavior and safety (*the egocentric perspective*). Intuitively, we seek more accurate descriptions of object geometry when it's more likely to interfere with the ego-agent's motion trajectory. However, current detection metrics, based on box Intersection-over-Union (IoU), are object-centric and are not designed to capture the spatio-temporal relationship between objects and the ego-agent. To address this issue, we propose a new egocentric measure to evaluate 3D object detection: Support Distance Error (SDE). Our analysis based on SDE reveals that the egocentric detection quality is bounded by the coarse geometry of the bounding boxes. Given the insight that SDE can be improved by more accurate geometry descriptions, we propose to represent objects as amodal contours, specifically amodal star-shaped polygons, and devise a simple model, StarPoly, to predict such contours. Our experiments on the large-scale Waymo Open Dataset show that SDE better reflects the impact of detection quality on the ego-agent's safety compared to IoU; and the estimated contours from StarPoly consistently improve the egocentric detection quality over recent 3D object detectors.

## 1 Introduction

3D object detection is a key problem in robotics, including popular applications such as autonomous driving. Common evaluation metrics for this problem, e.g. *mean Average Precision (mAP)* based on *box Intersection-over-Union (IoU)*, follow an object-centric approach, where errors on different objects are computed and aggregated without taking their spatiotemporal relationships with the ego-agent into account. While these metrics provide a good proxy for downstream performance in general scene understanding applications, they have limitations for egocentric applications, e.g. autonomous driving, where detections are used to assist navigation of the ego-agent. In these applications, detecting potential collisions on the ego-agent's trajectory is critical. Accordingly, evaluation metrics should focus more on the objects closer to the planned trajectory and to the parts/boundaries of those objects that are closer to the trajectory.

Recent works have introduced a few modifications to evaluation protocols to address these issues, e.g., breaking down the metrics into different distance buckets [53] or using learned planning models to reflect detection quality [34]. However, they are either very coarse [53] or rely on optimized neural networks [34], making it difficult to interpret and compare results in different settings. In this

---

[*]Correspondence to bydeng@waymo.com

35th Conference on Neural Information Processing Systems (NeurIPS 2021), virtual.

paper, we take a novel approach to 3D object detection from an *egocentric perspective*. We start by reviewing the first principle: the detection quality relevant to the ego-agent's planned trajectory, both at the moment and in the future, has the most profound impact on the ability to facilitate navigation. This leads us to transform detection predictions into two types of distance estimates relative to the ego-agent's trajectory — *lateral distance* and *longitudinal distance* (Fig. 1). The errors on these two distances form our *support distance error (SDE)* concept, where the components can either be aggregated as the max distance estimation error or used independently, for different purposes.

Compared to IoU, SDE (as a shape metric) is conditioned on the spatio-temporal relationship between the object and the ego-agent. Even a small mistake in detection near the ego-agent's planned trajectory can incur a high SDE (as in Fig. 2 left, object 3). Additionally, SDE can be extended to evaluate the impact of detections to the ego-agent's future plans (for cases where an object comes close to the planned trajectory later in time). This is not feasible for IoU, which is invariant to the ego-agent position or trajectory(shown in Fig. 2).

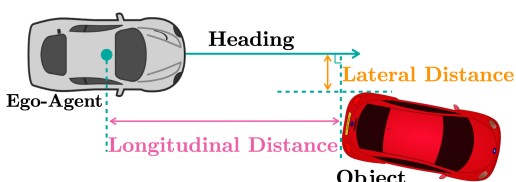

Figure 1: **Lateral distance and longitudinal distance.** These two types of support distance measure how far an object's *shape boundary* is to the observer (ego-agent) in both the direction along the observer velocity (longitudinal) and perpendicular to it (lateral).

Using SDE to analyze a state-of-the-art detector [44], we observe a significant error discrepancy between using a rectangular-shaped box approximation and the actual object's boundary, suggesting the need for a better representation to describe the fine-grained geometry of objects. To this end, we propose a simple lightweight refinement to box-based detectors named *StarPoly*. Based on a detection box, StarPoly predicts an amodal contour around the object, as a star-shaped polygon.

Moreover, we incorporate SDE into the standard average precision (AP) metric and derive an SDE-based AP (SDE-AP) for conveniently evaluating existing detectors. In order to make an even more egocentric AP metric, we further add inverse distance weighting to the examples, obtaining SDE-APD (D for distance weighted). With the proposed metrics, we observe different behaviors among several popular detectors [41, 44, 67, 20] compared to what IoU-AP would reveal. For example, PointPillars [20] excels on SDE-AP in the near range in spite of its less competitive overall performance. Finally, we show that StarPoly consistently improves upon the box representation of shape based on our egocentric metric, SDE-APD.

## 2  Related Work

**3D Object Detection**  Modern LiDAR-based 3D object detectors can be organized into three sub-categories based on the way they represent the input point cloud: *i.e.,* voxelization-based detectors [55, 8, 21, 50, 19, 60, 47, 68, 58, 20, 64, 56], point-based methods [45, 63, 32, 38, 62, 46] as well as hybrid methods [67, 61, 5, 12, 44]. Besides input representation, aggregating points across frames [13, 65, 14, 41], using additional input modalities [19, 4, 39, 57, 25, 29, 48, 37], and multi-task training [27, 59, 30, 24] have also been studied to boost the performance. Despite such progress in model design, the output representation and evaluation metrics have remained mostly unchanged.

**Egocentric Computer Vision**  Egocentric vision has been studied in various applications. To name a few, understanding human actions from egocentric cameras, including action/activity recognition [9, 35, 28, 49, 36, 51, 15, 52], action anticipation [43, 16], and human object interaction [26] have been widely studied. Egocentric hand detection/segmentation [23, 22, 1, 42], and pose estimation [54, 66, 31] are among other applications. Arguably, 3D detection for autonomous driving can be naturally viewed as another egocentric application where data is captured by sensors attached to the car. However, classic IoU-based evaluation metrics ignore the egocentric nature of this application.

**3D Object Detection Metrics**  Various extensions to the average precision (AP) metric have recently been proposed for the autonomous driving domain. nuScenes[2] consolidated mAP with more fine-grained error types. Waymo Open Dataset[53] introduced mAP weighted by heading (mAPH) to reflect the importance of accurate heading prediction in motion forecasting. [34] proposed to examine

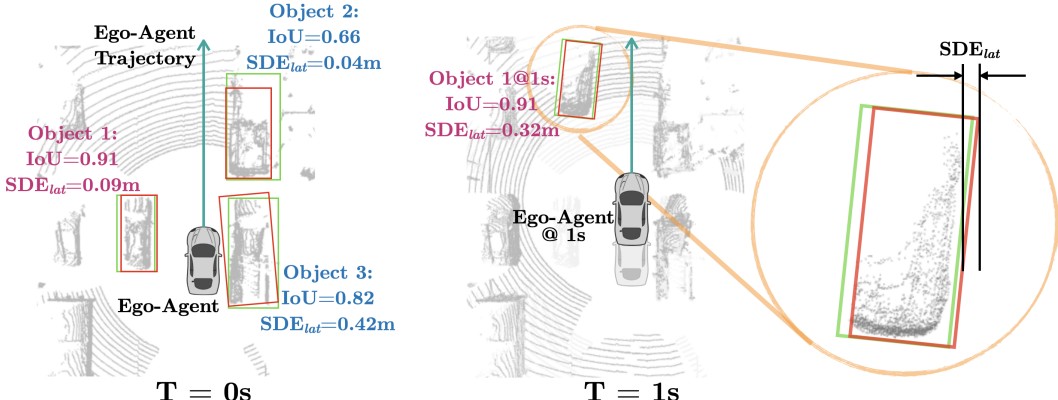

$T = 0s$          $T = 1s$

Figure 2: **Illustration of IoUs and lateral distances in a real scene.** We visualize the scene from a bird's eye view where Lidar points are **gray**; **green** boxes are ground truth boxes; and **red** boxes are detector boxes. *Left:* We show that IoU as an object-centric measure is not directly reflecting the risk of collision (colored in **blue**) — the high risk mistake of the object 3's box is not reflected by the high IoU. In contrast, while object 2 has a lower IoU, its box boundary is accurately estimated, thus the impact to ego-agent planning is limited. As shown, compared to IoU, SDE is more indicative of the perception quality's impact on driving. *Right:* We show how SDE changes when evaluated at a future time (colored in **purple**), reflecting how the current frame's perception quality influences decision making into the future. The detection box is transformed to a future frame based on the rigid motion between the ground truth boxes at $T = 0s$ and $T = 1s$ (which excludes the error introduced by motion prediction). While object 1 has low $\text{SDE}_{lat}$ at $T = 0s$ on the left, its error significantly increases at $T = 1s$, as the box cannot capture the fine-grained geometry at the object corner (see the zoom in view).

| Measure | TP Collision | | FP/FN Collision | |
|---------|------|--------|------|--------|
| | Mean | Median | Mean | Median |
| IoU ↑ | 0.903 | 0.912 | 0.904 | 0.903 |
| SDE ↓ | 0.114 | 0.094 | 0.162 | 0.153 |

Table 1: **Distributions of error measures in two types of collision detection cases.** In "TP Collision", both the ground truth points and the prediction report a collision. In "FP/FN Collision", either the ground truth (FN) or the prediction (FP) reports a collision. While the distributions of IoU in TP and FP/FN are close with even higher mean IoU in FP/FN, SDEs among TP are clearly better than FP/FN with an improvement of 30% in mean and 40% in median.

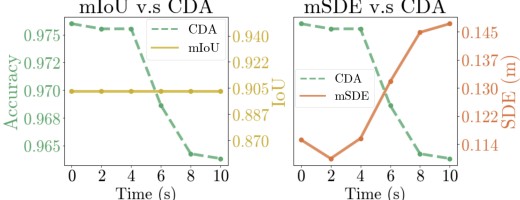

Figure 3: **Correlations with collision detection accuracy (CDA).** For each evaluation moments from the prediction time to $10s$ in the future, we compute the CDA, mean IoU (mIoU), and mean SDE (mSDE). We see from the curve that mIoU is not correlated with the accuracy drop as IoUs don't vary with ego motions; while mSDE is inversely correlated to collision detection accuracy due to its egocentric nature.

detection quality from the planner's perspective, by measuring the KL-divergence between future predictions conditioned on either noisy perception or ground truth. However, factors such as different planning algorithms or model training setups may cause this approach to yield inconsistent outcomes.

**Boundary-based Segmentation Metrics** A different class of shape metrics on semantic segmentation masks evaluates the match quality of ground truth and predicted segmentation *boundaries*. Representative methods include Trimap IoU[3, 18], F-score[10, 33], boundary IoU[6] etc. These methods operate in an object-centric manner and do not take temporal information into consideration.

## 3 An Egocentric Shape Metric: Support Distance Error

Understanding the quality of modern 3D object detectors from an egocentric perspective is an underexplored topic and is open for new egocentric shape measures. In this section, we first look at the limitations of the box Intersection-over-Union (IoU) measure, the de facto choice to evaluate detection quality in popular benchmarks [11, 53, 2] and then introduce our newly propose egocentric shape metric: support distance error (SDE).

**Limitations of box-based IoU**    IoU is an object-centric measure based on volumes (or areas). As illustrated in Fig. 2 (left), a prediction box with a relatively high IoU can still exhibit a high risk for an ego-agent (the protruding box can cause the planner to brake suddenly, which in turn could lead to a tailgating collision).

To understand such behavior at scale, we use collision detection as a "gold standard" to quantitatively reveal the limitation of IoUs. We select all the collisions reported by either the ground truth or a state-of-the-art PV-RCNN detector [44] in the validation set of the Waymo Open Dataset [53]. A ground truth collision is defined as an event where the object shape (approximated by the aggregated object LiDAR points across all of its observations) overlaps with the extended ego-agent shape (approximated by a bounding box of the ego-agent, scaled up by 80%). Collisions are estimated using detector boxes as the object's shape. Table 1 presents the mean and median IoUs for true positive and false positive collision detections, whose difference is minimal, indicating that IoU is not effectively reflecting collision risk.

**Support Distance Error (SDE)**    In autonomous driving, one of the core uses of detection is to provide accurate object distance and shape estimates for motion planning (which has collision avoidance as a one of the primary objectives). Instead of using box IoU, we can measure distances from the estimated shapes to the ego-agent's planned trajectory. Specifically, we propose two types of distance measurements (Fig. 1) [2]:

- **Lateral distance** to an object: The minimal distance from any point on the object boundary to the line in the ego-agent's heading direction. This distance is critical for the ego-agent to plan lateral trajectory maneuvers.
- **Longitudinal distance** to an object: The minimal distance from any point on the object boundary to the central line perpendicular to the ego-agent's heading direction. This distance is important to determine the speed profile and keep a safe distance from the objects in front.

We use the term *support distances* for these two distance types, as they "support" the decision making in trajectory planning, and name the error between the ground truth support distance and the one estimated from a detector's output as the *support distance error (SDE)*. We use $SDE_{lat}$ to denote the lateral distance error and $SDE_{lon}$ for the longitudinal error, and we define SDE as the maximum of the two. This formulation leads to two conceptual changes compared to IoU: we shift our focus from *volume* to *boundary* and from *object-centric* to *ego-centric*.

This definition can also be extended to measure the impact of the detection quality on *future* collision risks. If we compute distances from the object boundary to the tangent lines at a future position (at time $t$) on the ego-agent's trajectory, we can compute SDE for different future time steps (denoted as SDE@t). This is equivalent to measuring how close the object is to a future location of the ego-agent.

To make the definition concrete, at time $T = t$, we assume the ego-agent's pose is $e^{(t)} = (x^{(t)}, \theta^{(t)})$, with $x^{(t)} \in \mathcal{R}^3$ as its center (e.g. the center of the ego-agent's bounding box) and $\theta^{(t)}$ as its heading direction (e.g. clock-wise rotating angle around the up-axis). We define the "lateral line", the line crossing the ego-agent's center and in the direction of its heading, as $l_{lat}^{(t)}$; and the "longitudinal line" perpendicular to it as $l_{lon}^{(t)}$. On the other hand, we assume we have an object $o$ and its predicted boundary is $B(o)$ as a set of points on the boundary. The lateral/longitudinal distance of $o$ at the current frame ($T = 0$) is defined as:

$$SD_\alpha = SD_\alpha(B(o), e^{(0)}) = \min_{p \in B(o)} d(p, l_\alpha^{(0)}), \alpha \in \{lat, lon\} \tag{1}$$

where $d$ computes the point-to-line distance. If the line passes through the object boundary the SDE would be 0. Assume $B_{gt}(o)$ is the object ground truth boundary, then the lateral/longitudinal support distance error is defined as:

$$SDE_\alpha = SD_\alpha(B_{gt}(o), e^{(0)}) - SD_\alpha(B(o), e^{(0)}), \alpha \in \{lat, lon\} \tag{2}$$

The SDE sign has a physical meaning: positive errors mean the predicted boundary is protruding while negative means that a part of the object is not covered by the predicted boundary. For

---

[2]For simplicity, we define the distances from the object boundary to the ego-agent trajectory (or the line perpendicular to it), instead of using the ego-agent shape, which varies across datasets and is typically not available in public datasets.

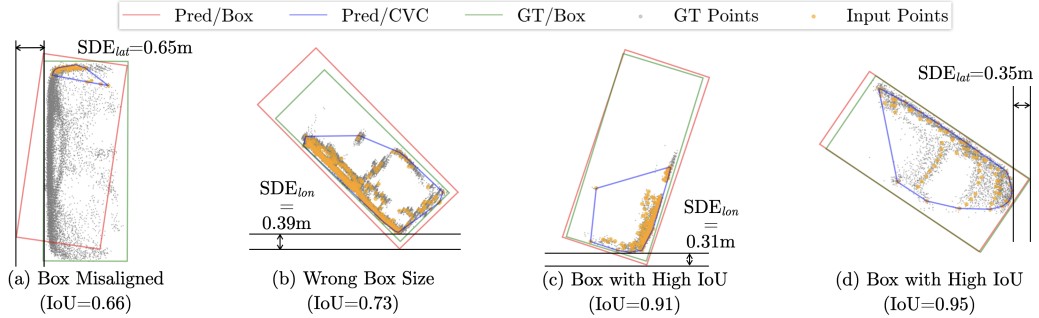

Figure 4: **Failure cases with large SDEs** ($>= 0.3m$). (a) and (b): The detector boxes are poorly aligned with the ground truth either in orientation or size. (c) and (d): The detector boxes yield near-perfect IoUs with the ground truths but still incur high SDE. The convex visible contours (CVC) are derived based on the input points within a detection at the current frame. Note that SDEs here are computed against temporally aggregated Lidar points (the GT Points) and IoUs are computed between detections and ground truth boxes.

simplicity, we take the absolute value of $SDE_{lat}$ and $SDE_{lon}$ by default and formally define $SDE = \max(|SDE_{lat}|, |SDE_{lon}|)$, an aggregated value of both errors.

To measure the impact of current frame detection quality on future plans, we define SDE@t, which computes the SDE of an object $t$ seconds in the future. Given the ground truth rigid motion $R^{(t)}$ of the object from $T = 0$ to $T = t$, we can transform its predicted boundary at frame $T = 0$ to its future position. In this way, the error patterns of the boundary can be consistently propagated into a future frame (see Fig. 2 right for an example). The rigid motion can be derived between pairs of ground truth boxes of the object. We denote the transformed $B(o)$ as $B^{(t)}(o)'$. Note that it is different from the object shape prediction at time $T = t$: we are still measuring the quality of the $T = 0$ prediction, but within a future egocentric context. The future support distance can be formally defined as:

$$SD_\alpha@t = SD_\alpha(B^{(t)}(o)', e^{(t)}) = \min_{p \in B^{(t)}(o)'} d(p, l_\alpha^{(t)}), \alpha \in \{lat, lon\} \tag{3}$$

Similarly, we define $SDE_\alpha@t$ as the difference in $SD_\alpha@t$ between the ground truth and the predicted boundary, where $\alpha \in \{lat, lon\}$. Then $SDE@t = \max(|SDE_{lat}@t|, |SDE_{lon}@t|)$. We use SDE and SDE@0s interchangeably unless otherwise noted.

**Metric implementation details**   To faithfully compute the support distance, we aggregate object surface points (from Lidar) across all frames, during which the object is observed (which cover different viewpoints of the object) as a surrogate shape to the ground truth. This allows us to effectively compute distances to the boundary without requiring costly object shape annotations/modeling. By default, we use the real driving trajectory. The same implementation applies when one would like to evaluate SDE by providing an arbitrary set of intended trajectories (from a planner or simulation).

**Comparing SDE with IoU**   In Fig. 2, we see $SDE_{lat}$ is a highly useful indicator to reflect collision risk (for object 3). In Tab. 1, we show that the mean and median SDE are sensitive shape measures and are inversely correlated with the collision risk. Naturally with larger $t$, SDE@t increases, since the detections are based only on sensor data from the current frame $T = 0$. Fig. 3 shows how SDE@t and IoU change when we evaluate them at different time steps.

Note that both SDE and SDE@t are defined based on distances to the object boundary (usually the part closer to the ego-agent). Clearly, better detection quality and boundary representation will result in an improved SDE metrics, which leads to the main idea of our next section.

## 4   Shape Representations and SDE

In this section, we use SDE to analyze detection quality in safety-critical scenarios and highlight the importance of the shape representation therein. We further propose a new amodal contour representation and a neural network model (StarPoly) for contour estimation and demonstrate it produces significant SDE improvements.

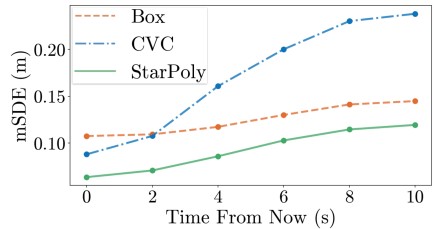

|  | PV-RCNN | | | Ground Truth | | |
|---|---|---|---|---|---|---|
|  | Box | CVC | StarPoly | Box | CVC | StarPoly |
| mSDE of $[0m, 5m)$ | 0.107 | 0.090 | 0.063 | 0.059 | 0.083 | 0.046 |
| mSDE of $[5m, 10m)$ | 0.108 | 0.087 | 0.064 | 0.070 | 0.076 | 0.053 |
| mSDE of $[10m, 20m)$ | 0.140 | 0.155 | 0.086 | 0.094 | 0.142 | 0.068 |
| mSDE of $[20m, 40m)$ | 0.207 | 0.266 | 0.152 | 0.132 | 0.235 | 0.105 |

Table 2: **Comparing mean SDE (mSDE) of boxes, convex visible contours (CVC), and our StarPoly** at different distance ranges. While lower than mSDE of detector box, CVC's SDE rises rapidly towards far ranges. Meanwhile, StarPoly is superior than both box and CVC in all ranges.

Figure 5: **mSDE in $[0m, 10m)$ at different time steps.** CVC's mSDE significantly increases as the evaluation goes into the future. In contrary, StarPoly consistently outperforms others.

## 4.1 Qualitative Analysis of Bounding Box Failure Cases

To understand how detector boxes perform under the SDE measure, we select PV-RCNN [44], a top-performing single-frame point-cloud-based detector in popular autonomous driving benchmarks [11, 53]), for our analysis. All analysis is based on the Waymo Open Dataset [53] validation set. Fig. 4 illustrates some representative failure cases among the detector boxes, with high SDE.

We find that even when a box aligns reasonably well with the ground truth it can still incur high SDE. By comparing the predicted detection against the point cloud inside in the box, we notice that rectangular boxes typically do not tightly surround the object boundary. In particular, the discrepancy between box corners and the actual object boundary contributes a considerable amount of SDE. This observation inspires us to seek more effective representations of the fine-grained object geometry.

## 4.2 Convex Visible Contours

An intuitive solution to obtain a tighter object shape fit is by leveraging the Lidar points. Specifically, one can extract all points within the detector box (after removing points on the ground) and compute their convex hull, as a convex visible contour (CVC). In contrast to amodal object shape, CVC is computed only from the visible Lidar points at the current frame. Fig. 4 provides some visualizations.

Tab. 2 shows how CVC compares with bounding boxes in SDE. Considering that CVC is heavily dependent on the quality of the box it resides in, we also evaluate CVC directly based on the ground truth boxes, which can be seen as the upper bound for CVC (col. 6). We see that at near range, CVC can significantly improve SDE compared to the detector boxes (col. 2 vs 3). However, its effectiveness degrades at longer ranges (col. 2 vs 3) and its performance is inferior to ground truth boxes (col. 5 vs 6). We hypothesize that this is because CVC is vulnerable to occlusions, clutter and object point cloud sparsity at longer ranges, which are ubiquitous phenomena in real world data. In Fig. 5, the analysis based on SDE@t confirms that CVC performs better than detector boxes at the current frame but generalizes poorly to longer time horizons. To improve it, we need a representation that provides good coverage of both the visible and the occluded object parts.

## 4.3 Amodal Contour Estimation with StarPoly

We propose to refine box-based detection with *amodal contours*, a polyline contour that covers the entire object shape (See Fig. 6 for an illustration). Our model, StarPoly, implements contours as star-shaped polygons and predicts amodal shape via a neural network [3]. It can be employed to refine predicted boxes for any off-the-shelf detectors.

**Input**    The input to the StarPoly model is a normalized object point cloud. We crop the object point cloud from its (extended) detection box. The point cloud is canonicalized based on the center and the heading of the detection box, as well as scaled by a scaling factor, $s$, such that the length of the longest side of the predicted box becomes 1.

---

[3]Although there are previous works on shape reconstruction/completion [7, 30], they are often trained on synthetic data and are not directly applicable to real Lidar data. We leave more studies in designing the best contour estimation model to future work and evaluate StarPoly as a baseline towards better egocentric detection.

**Parameterization**   As shown in Fig. 6, the star-shaped polygon is defined by a center point, $h$, and a list of vertices on its boundary, $(v_1, ..., v_n)$, where $n$ is the total number of vertices determining the shape *resolution*. We assume $h$ is the center of the predicted box and sort $(v_1, ..., v_n)$ in clockwise order so that connecting the vertices successively produces a polygon. We constrain $v_i$ to have only 1 degree of freedom by defining $v_i = c_i \vec{d_i}$, where $(\vec{d_1}, ..., \vec{d_n})$ is a list of unit vectors in predefined directions. Consequently, predicting a star-shaped polygon is equivalent to predicting $(c_1, ..., c_n)$, for which we employ a PointNet [40] model (see the supplementary material for details).

**Optimization**   Since ground truth contours are not available in public datasets, directly training the regression of $(c_1, ..., c_n)$ is infeasible. We resort to a surrogate objective for supervision. The objective combines three intuitive goals, namely *coverage*, *accuracy*, and *tightness*. The coverage loss encourages the prediction to encompass all ground truth object points (aggregated points in the object bounding box from all frames in which the object appears, with ground points removed). Moreover, as the input point cloud already reveals part of the object boundary visible to the ego-agent, the accuracy loss requires the prediction to fit the visible boundary as tight as possible. On the other hand, the tightness loss minimizes the area of the predicted contour. The combination of these three goals leads to the reconstruction of contours without requiring ground truth contour supervision. More formally, the coverage loss $\mathcal{L}_c$, the accuracy loss $\mathcal{L}_a$, the tightness loss $\mathcal{L}_t$, and consequently the overall objective $\mathcal{L}$ for one ground truth point cloud $X$ are defined as follows:

$$
\begin{aligned}
\mathcal{L} = &\frac{1}{|X|} \sum_{x \in X} \max \left( \frac{x \times v_r}{v_l \times v_r} + \frac{v_l \times x}{v_l \times v_r} - 1, 0 \right) && \implies \text{Encompass all object points, } \mathcal{L}_c. \\
&+ \beta \frac{1}{|B|} \sum_{x \in X} \left| \frac{x \times v_r}{v_l \times v_r} + \frac{v_l \times x}{v_l \times v_r} - 1 \right| && \implies \text{Fit tight to visible boundaries, } \mathcal{L}_a. \quad (4) \\
&+ \gamma \frac{1}{n} \sum_i \|c_i\| && \implies \text{Minimize the area of contours, } \mathcal{L}_t.
\end{aligned}
$$

where $x$ is a point from $X$, $\gamma$ is a weight parameter for $\mathcal{L}_t$, and $\times$ represents cross product. Note that in $\mathcal{L}_c$, $l$ and $r$ are selected so that $\vec{d_l}$ and $\vec{d_r}$ span a wedge shape containing $x$ (as shown in Fig. 6). Intuitively, $\mathcal{L}_c$ is computing the barycentric coordinates of $x$ with regard to $v_l$ and $v_r$ within the triangle $\triangle h v_l v_r$ and encouraging $x$ to be on the same side as $h$ regarding $v_l v_r$ — the necessary and sufficient condition for $x \in \triangle h v_l v_r$. Similarly, $\mathcal{L}_a$ is forcing the points on the visible boundary $B$ to be on the predicted boundary as well. Meanwhile, $\mathcal{L}_t$ is pulling all $v_i$ towards $h$.

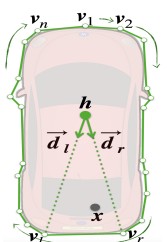

Figure 6: StarPoly formulation.

**Results**   In Tab. 2 we see that updating the bounding box output of PV-RCNN to StarPoly contours significantly improves the mean SDE under all distance buckets (e.g. at 0-5m, it improves from 10.7cm to 8.6cm, which is around 20% error reduction). Similar improvements also appear on the ground truth boxes (col. 4-6). In Fig. 5, we also show how StarPoly improves on SDE@t. In all time steps, the StarPoly has lower SDE than both bounding boxes and visible contours, showing its advantage of getting the best of both worlds.

## 5   Egocentric Evaluation of 3D Object Detectors

In this section, we incorporate SDE into the standard average precision (AP) metric and evaluate various detectors and shape representations on the Waymo Open Dataset [53].

**SDE-AP: Detection AP based on the SDE shape metric**   To compare different detectors on their egocentric performance, we cannot just use the SDE measure, which does not consider false positive (FP) and false negative (FN) cases. Therefore, we propose to adapt the traditional IoU-based AP (IoU-AP) to an SDE-based one (SDE-AP). Specifically, we replace the classification criterion for true positives (TP) from an IoU to an SDE-based threshold and use SDE = 20cm as the threshold (see the

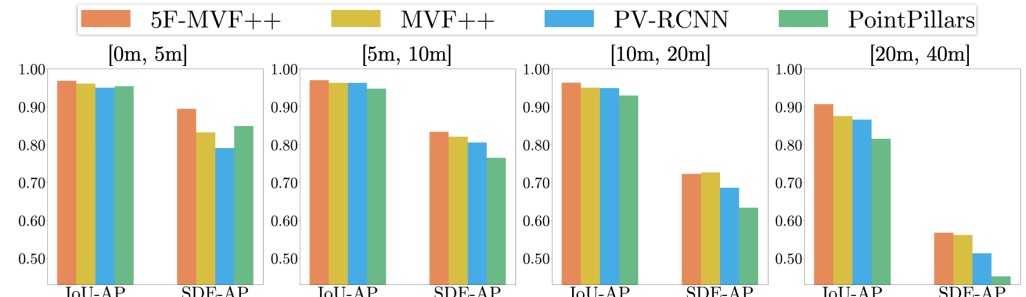

Figure 7: **Distance breakdowns of IoU-AP and SDE-AP.** SDE-AP can better differentiates different egocentric detection quality than IoU-AP, especially in the near ranges ($[0m, 5m]$ and $[5m, 10m]$).

Supplementary material for more on why we selected this number). In addition, we use SDE-based criterion (instead of an IoU-based one) to match predictions and ground truth.

**SDE-APD: inverse distance weighted SDE-AP** Although SDE-AP is based on the egocentric SDE measure, it weights objects at various distances from the ego agent trajectory equally. To design a more strongly egocentric measure, we further propose a variant of the SDE-AP with inverse-distance weighting, termed SDE-APD (the suffix D means distance weighted). Specifically, for a given frame we have detections $B = \{b_i\}, i = 1, ..., N$ and ground truth objects $G = \{g_j\}, j = 1, ..., M$. We denote the matched ground truth object for $b_i$ as $g(b_i) \in G$. A prediction is counted as a true positive if $\text{SDE}(b_i, g(b_i); e) < \delta$ where $\delta$ is the SDE threshold and $e$ is the ego-agent pose. Then we define the set of true positive predictions as $TP = \{b_i | SDE(b_i, g(b_i); e) < \delta\}$ and false positive predictions as $FP = B - TP$. The inverse distance weighted TP count (IDTP), FP count (IDFP) and ground truth count (IDG) for the frame are:

$$IDTP = \sum_{b_i \in TP} 1/d_{g(b_i)}^{\beta} \quad IDFP = \sum_{b_i \in FP} 1/d_{b_i}^{\beta} \quad IDG = \sum_{g_i \in G} 1/d_{g_i}^{\beta} \quad (5)$$

where $d$ is the Manhattan distance from the prediction shape center to the ego-agent center and $\beta$ is a hyper parameter controlling how much we focus on the close-by objects (we set $\beta = 3$, see the supplementary for more details).

The inverse distance weighted precision and recall are defined as $IDTP/(IDTP + IDFP)$ and $IDTP/IDG$ respectively, both remain within $[0, 1]$. The SDE-APD is the area under the PR-curve. Similar to SDE, which is defined both for the current frame and for future frames (SDE@t), the SDE-AP and SDE-APD metric also have future equivalents SDE-AP@t and SDE-APD@t that can evaluate the impact of current frame perception on future plans.

### 5.1 Comparing Different Detectors on SDE-AP and SDE-APD

In this subsection, we compare a few representative point-cloud-based 3D object detectors on the SDE-AP and SDE-APD metrics. We study several popular detectors: PointPillars [20], a light-weight and simple detector widely used as a baseline; PV-RCNN [44], a state-of-the-art detector with a sophisticated feature encoding; MVF++ [67, 41] (an improved version of the multi-view fusion detector), a recent top-performing detector; and finally 5F-MVF++ [41], an extended version of MVF++ taking point clouds from 5 consecutive frames as input, the most powerful among all.

| Method | SDE-APD | IoU-AP |
|---|---|---|
| 5F-MVF++ | 0.874 | 0.863 |
| MVF++ | 0.834 | 0.814 |
| PV-RCNN | 0.808 | 0.797 |
| PointPillars | 0.817 | 0.720 |

Table 3: **SDE-APD and IoU-AP[4] of different detectors.**

Fig. 7 shows the SDE-AP with distance breakdowns for all detectors and Table 3 shows the egocentric SDE-APD metric. An interesting observation from Fig. 7 about IoU-AP is that, while the four detectors have fairly close IoU-APs at close ranges (e.g. [0m, 5m]), we see significant gaps among them at longer ranges (e.g. [20m, 40m]). Since there are more objects at longer ranges, those long-range buckets typically dominate the overall IoU-AP. In contrast, the SDE-AP is consistently more discriminative of the detectors especially for the very short range

---

[4]The IoU-AP is compute using euclidean distance matching and 2D IoU 0.7 as the threshold.

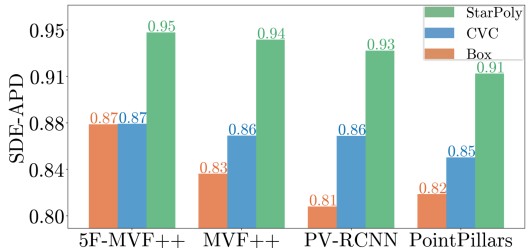

Figure 8: **SDE-APD of detectors with different output representations (T=0s).**

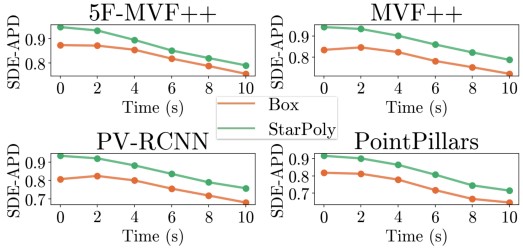

Figure 9: **SDE-APD@t of boxes and StarPoly based on different detectors.**

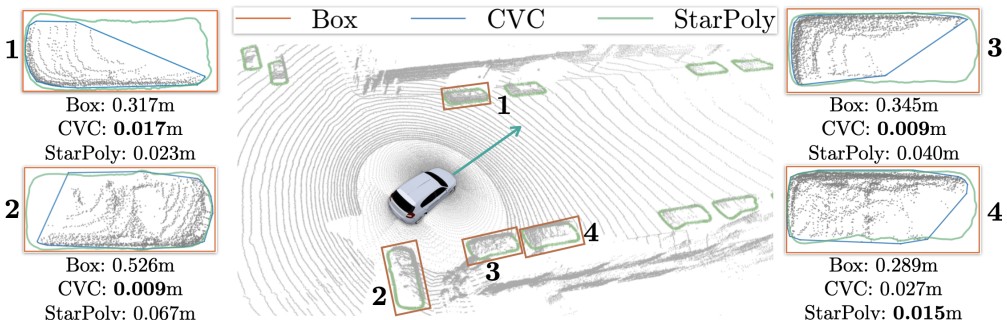

Figure 10: **Qualitative Results.** A scene from the validation set of the Waymo Open Dataset with StarPoly predictions shown as green contours. We also zoomed in into 4 vehicles closest to the ego-agent and compared StarPoly (**green**) with predicted box (**red**), and CVC (**blue**). $SDE_{lat}$ are reported under each zoom-in.

in [0m, 5m]. We even see some change of rankings – PointPillars, with the lowest overall IoU-AP, outperforms PV-RCNN and MVF++ in close-range [0m, 5m] SDE-AP, suggesting it has a particularly strong short-range performance. This also implies that simply examining IoU-AP for selecting detectors can be sub-optimal and our SDE-AP can provide an informative alternative perspective.

## 5.2 Comparing Various Shape Representations

In this subsection, we evaluate how detector output representations affect the overall detection performance in terms of SDE-APD evaluated at the current frame as well as into the future.

**StarPoly implementation details.** For the encoding neural network, we use the standard Point-Net [40] architecture followed by a fully-connected layer to transform latent features to $(c_1, ..., c_n)$. We use a resolution $n = 256$ for all following experiments. $(\vec{d}_1, ..., \vec{d}_n)$ is uniformly sampled from the boundary of a square. During training, $\gamma$ and $\beta$ are both set to $0.1$, which is determined by a grid search over the hyperparameters. Please refer to the supplementary material for more model details.

**Results** In Fig. 8, we compare the egocentric performance of different representations using SDE-APD. StarPoly consistently improves the egocentric result quality across the different detectors. Interestingly, StarPoly largely closes the gap between the different detectors, reducing the difference between 5F-MVF++ [41] and PV-RCNN by a factor of 3. This implies that StarPoly's amodal contours can greatly compensate for the limitations of the initial detection boxes, especially of those with poorer quality. StarPoly also outperforms convex visible contours (CVC) across all detectors. In Fig. 9, we evaluate the performance of the different representations at future time steps. We observe that StarPoly remains superior to the detector boxes across all time steps, differentiating it from the convex visible contours that decay catastrophically over time (shown in Fig. 5). Fig. 10 shows a scene with StarPoly amodal contours estimated for all vehicles. The zoom-in figures reveal how amodal contours have more accurate geometry, and lower SDE, than both boxes and visible contours. However, they are not yet perfect, especially on the occluded object sides. Improving contour estimation even further is a promising direction for future work.

## 6    Conclusion

In this paper, we propose egocentric metrics for 3D object detection, measuring its quality in the current time step, but also its effects on the ego agent's plans in future timesteps. Through analysis, we have shown that our egocentric metrics provide a valuable signal for robotic motion planning applications, compared to the standard box intersection-over-union criterion. Our metrics reveal that the coarse geometry of bounding boxes limits the egocentric prediction quality. To address this, we have proposed using amodal contours as a replacement to bounding boxes and introduced StarPoly, a simple method to predict them without direct supervision. Extensive evaluation on the Waymo Open Dataset demonstrates that StarPoly improves existing detectors consistently with respect to our egocentric metrics.

**Acknowledgements and Funding Transparency Statement.**    We thank Johnathan Bingham and Zoey Yang for the help on proofreading our drafts, and the anonymous reviewers for their constructive comments. All the authors are full-time employees of Alphabet Inc. and are fully-funded by the subsidairies of Alphabet Inc., Google LLC and Waymo LLC. All experiments are done using the resources provided by Waymo LLC.

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
