# Revisiting 3D Object Detection From an Egocentric Perspective

## Supplementary Material

This document provides supplementary content to the main paper. In Sec. A, we expand the discussion on comparing our egocentric metric with a recent planner-based 3D object detection metric. In Sec. B, we provide more details of the StarPoly architecture and training. Sec. C explains more about how we select hyperparameters for our egocentric metrics. Finally, Sec. D shows more visualization results.

## A  Discussion

Similar to our metric, a recent work (planner-centric metrics) [34] also follows an egocentric approach to evaluate 3D object detection. It measures the KL-divergence of the planner's prediction based on either the ground truth or the detection. However, we would like to highlight two differences between our SDE-based metrics (SDE-AP and SDE-APD) and the planner-centric metrics: *stability* and *interpretability*.

**Stability.**  In the planner-centric metrics [34], a pre-trained planner is required for the evaluation. Consequently, the metric is highly dependent on the architectural choices of the planner and may vary drastically when switched to a different one. Moreover, as the proposed planner is learned from data, many factors in the training can significantly affect the evaluation outcome: 1) the metric depends on a stochastic gradient descent (SGD) optimization to train the planner, which may fall into a local minimum; 2) the metric depends on a training set of trajectories, which will vary depending on the shift of data distribution. Furthermore, if the planner is trained on ground truth boxes, it may not reflect the preferences of a practical planner which is usually optimized for a certain perception stack. In contrast, SDE-based metrics don't require any parametric models. Its evaluation is consistent and can be universally interpreted across different datasets or downstream applications.

**Interpetability.**  Because the KL-divergence employed in [34] only conveys the correlation of two sets of distributions, the magnitude of the metric is difficult to interpret. To fully understand the detection errors, one has to investigate the types of failures made by the planner, which vary depending on type of planner used. On the other hand, our proposed SDE directly measures the physical distance estimation error in meters. For SDE-based AP metrics, an intuitive interpretation is the frequency of detection, whose distance estimation error is within an empirically set threshold. Therefore, SDE-based metrics have a clear physical meaning, which translates the complex model predictions into safety-sensitive measurements.

## B  StarPoly Model Details

### B.1  Architecture

Our StarPoly model takes as input the point clouds cropped from (extended) detection bounding boxes. We apply a padding of $30cm$ along the length and width dimensions for for all detection boxes before the cropping. The point cloud is normalized before being fed into StarPoly based on the center, dimensions, and heading of each bounding box. In addition, the point cloud is subsampled to 2048 points before being processed by StarPoly. We use a PointNet [40] to encode the point cloud into a latent feature vector of 1024-d. Then, we reduce the dimensions of the latent feature vector from 1024-d to 512-d with a fully-connected layer. At last, another fully-connected layer is employed to predict the $n$-d parameters of a star-shaped polygon, where $n$ is the resolution of the star-shaped polygon (as stated in Sec. 4.3). We use $n = 256$ for all the experiments in the main paper. As for selecting $(\vec{d}_1, ..., \vec{d}_n)$, we uniformly sample directions on the boundary of a square, inspired by the prior that the objects of interest in this paper, i.e. vehicles, are symmetrical and are approximately of rounded square or rectangular shapes.

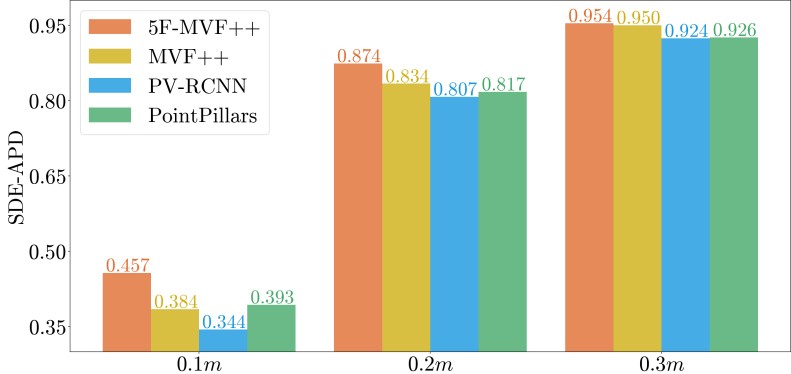

Figure 11: **SDE-APD with various distance thresholds.** Note that at more stringent error threshold, e.g., $0.1m$, SDE-APD clearly differentiates different detector's detected box quality, where 5F-MVF++ keeps outperforming others and PointPillars excels as well.

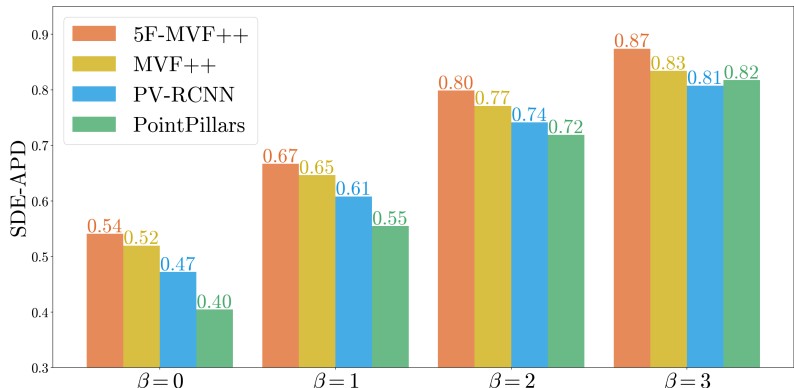

Figure 12: **SDE-APD with various $\beta$.** We change the inverse distance weighting degree, $\beta$, in SDE-APD computation. Note that as we increase the degree, which means more focus is shifted to close objects, the SDE-APD of PointPillars [20] gradually catches up and even surpasses PV-RCNN [44] at $\beta = 3$. This is coherent with our study on SDE-AP's distance breakdowns. We can therefore conclude that $\beta$ is a knob in SDE-APD to control the level of egocentricity based on object distances.

## B.2  Optimization

We train StarPoly on the training split of the large-scale Waymo Open Dataset [53]. Because StarPoly aims to refine the results of a detector, we first use a pre-trained detector to crop out point clouds as described in Sec. B.1. Then we optimize StarPoly *independently* using the prepared point clouds. For all the experiments in the paper, we use StarPoly trained on MVF++. We find that StarPoly can generalize to different detectors even if trained only on one detector. For the StarPoly optimization, we use the Adam optimizer [17] with $\beta_1 = 0.9$, $\beta_2 = 0.99$ and learning rate $= 0.001$ an the parameters. For all experiments in the paper, we train StarPoly for $500,000$ steps with a batch size of $64$ and set $\gamma = 0.1$.

## C  Details about SDE and SDE-APD

### C.1  Selection of metric hyperparameters

**Selection of the SDE threshold**  As defined in main paper Sec. 5, we classify *true positive* predictions by comparing the SDE with a threshold. We use a threshold of $20cm$ for all experiments in the paper. Unlike the IoU threshold, our threshold has a direct physical meaning in safety-critical scenarios, i.e. the amount of estimation error by perception that an autonomous vehicle can handle. Therefore, it can be selected according to the real world use cases. In this paper, we select $20cm$ via analyzing the SDE of ground truth bounding boxes (as shown in main paper Table 2). We find the

overall mean SDE of it to be $0.1m$ and therefore determine a relaxed value of $0.2m$ as the threshold. One can also use different thresholds for the evaluation as one can use different IoU thresholds for the box classification. Figure 11 illustrates the comparison among detectors with varying thresholds, *i.e.,* $0.1m$, $0.2m$, $0.3m$. We can see that PointPillars [20] demonstrates stronger performance compared to PV-RCNN [44] when the threshold is set more stringent. In addition, the effectiveness of using multi-frame information is more pronounced when the evaluation criterion becomes more rigorous.

**Selection of $\beta$ in the Inverse Distance Weighting**   To be more egocentric in our evaluation, we propose to extend the Average Precision (AP) computation by introducing inverse distance weighting. This strategy aims to automatically emphasize the objects close to the ego-agent's trajectory than those far away. As the number of objects grows roughly quadratically with regard to the distance, setting $\beta = 2$ (square inverse) would put equal weight for all distances. Since we want to highlight the importance of close-by objects, we go a further step and set $\beta = 3$.

Fig. 12 shows the SDE-APD (evaluated at time step 0) with different choices of $\beta$. Setting $\beta = 0$ means all objects contribute equally to the AP metric, where see the greatest gap from the best and the worst detectors (5F-MVF++ [41] v.s. PointPillars [20]). As we increase the $\beta$, i.e. making the overall AP metric more egocentric, weighting more heavily on the close-by objects, we see the PointPillars (with great close-by accuracy) catches up with PV-RCNN [44] and MVF++. The general differences of different detectors also become smaller as they perform similarly well for objects close to the ego-agent's trajectory (the difference in the original IoU-AP is more related to their performance difference on far-away objects).

### C.2    Importance of inverse distance weighting and SDE in SDE-APD

In SDE-APD, we introduce inverse distance weighting as a simple proxy of distance breakdowns. To investigate the impact of such weightings, we extend IoU-AP to IoU-APD with the same distance weighting as SDE-APD. The results are shown in Tab. 4. Note that while IoU-APD and IoU-AP have the same ordering, SDE-APD is able to reveal a different ranking between PointPillars and PV-RCNN, where we claim that SDE plays a more important role.

| Method | SDE-APD | IoU-APD | IoU-AP |
|---|---|---|---|
| 5F-MVF++ | 0.874 | 0.989 | 0.863 |
| MVF++ | 0.834 | 0.981 | 0.814 |
| PV-RCNN | 0.808 | 0.972 | 0.797 |
| PointPillars | 0.817 | 0.966 | 0.720 |

Table 4: **SDE-APD, IoU-APD, and IoU-AP of different detectors.**

### C.3    Composition of lateral and longitudinal distance errors in SDE

In our default definition, SDE is the maximum value of the lateral distance error and the longitudinal error. In Tab. 5 we investigate the composition of the two sub-distance-errors of the SDE. Specifically, we employ the detection boxes predicted by PV-RCNN as the detection output and calculate the mean and average of all valid $SDE_{lat}$ and $SDE_{lon}$. Note that "valid" means that the object doesn't intersect with the lateral line (for $SDE_{lat}$) or the longitudinal line (for $SDE_{lon}$) and that the box is matched with a ground truth object. We also compute the portion of SDEs that are equal to its lateral component, i.e. $SDE_{lat} > SDE_{lon}$, and the portion of SDEs that are equal to

| Statistics | $SDE_{lat}$ | $SDE_{lon}$ |
|---|---|---|
| Mean $(m)$ | 0.17 | 0.17 |
| Median $(m)$ | 0.12 | 0.11 |
| Contribution | 52% | 48% |

Table 5: **Composition of SDE from PV-RCNN's Detection Boxes.**

the longitudinal component. From the statistics, we find that the lateral and longitudinal components contribute almost equally to the final SDE.

### C.4    Distribution of signed SDE

In this work, we intend to bring attention to the idea of egocentric evaluation. We propose SDE without sign as a simple implementation of this idea with minimal hyperparameters required. It is straightforward to extend it to more complicated versions with the sign included. In Fig. 13, we provide a plotting of the distribution of signed SDE of detector boxes. It demonstrates that box predictions are generally oversized, i.e. with positive SDEs. Based on specific requirements of an

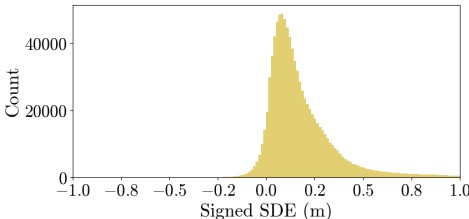

Figure 13: **Distribution of Signed SDE.** We show the distribution of $\max(\text{SDE}_{lat}, \text{SDE}_{lon})$ of PV-RCNN's box detections, where positive means over-sized predictions while negative means under-sized. Box predictions have an oversizing bias.

| Measure | TP Collision | | FP/FN Collision | |
|---|---|---|---|---|
| | Mean | Median | Mean | Median |
| IoU $\uparrow$ | 0.902 | 0.911 | 0.904 | 0.903 |
| SDE $\downarrow$ | 0.114 | 0.095 | 0.161 | 0.153 |

Table 6: **Distributions of error measures in two types of collision detection cases.** In "TP Collision", both the ground truth points and the prediction report a collision. In "FP/FN Collision", either the ground truth (FN) or the prediction (FP) reports a collision. Here we use the aggregated point clouds to test collision. The results align with Tab. 1.

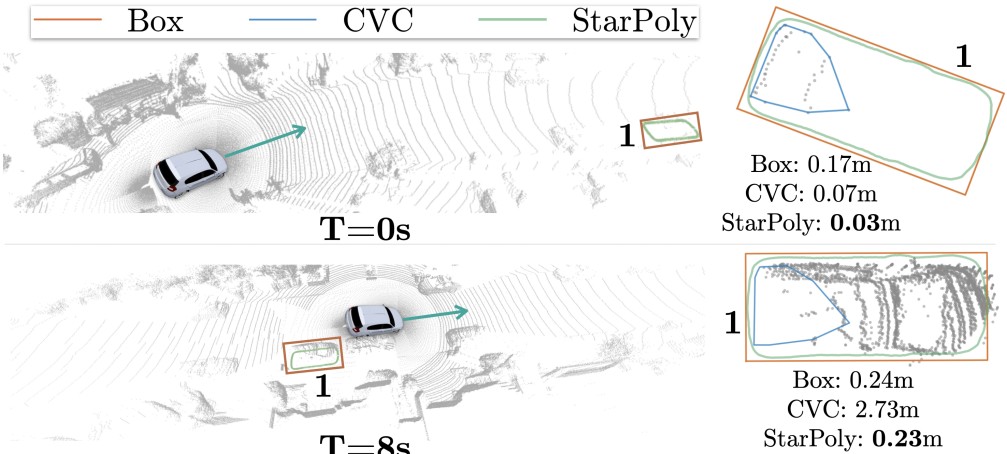

Figure 14: **Qualitative Results** for evaluating predictions at a future time step. Top: predictions at time T=0. Bottom: evaluations at T=8s. On the right of each row is the zoom-in view where the prediction and point cloud cropped by the ground truth bounding box are shown. $\text{SDE}_{lon}$s are reported under zoom-ins for each representation. A far away object at T=0 can become very close to the agent in a future time step (as shown for T=8s). While convex visible contour (CVC) may achieve comparable results to StarPoly at T=0, its performance considerably drops when evaluated at T=8s. This is why StarPoly achieves better results than the box and CVC representations across different time steps.

application, one can also have more fine-grained thresholds, e.g. different thresholds for positive and negative, and select the most suitable set up based on their priorities.

### C.5 Collision correlation of SDE and IoU based on contours

In Tab. 1, we use ground truth box to test collisions, to align with the evaluation of IoU. In Tab. 6, we re-computed the table using the contours drawn from our aggregated ground truth points, which should be the more accurate shape accessible. The gap between IoU and SDE is almost the same as the original Tab. 1 using boxes for collision tests.

## D  Qualitative Results

In this section we provide additional qualitative analysis. Fig. 14 shows how our metrics evaluate predictions at a future time step. We compare different representations both at the current time frame and at a future time frame. Our metrics are egocentric in the sense that they take into account the relative positions of the objects to the agent's trajectory in both the current and *future* time steps. Clearly, our proposed representation, StarPoly outperforms both box and convex visible contour (CVC) representations at the future time step. Fig. 15 shows a case when the CVC fails to capture the full shape of the object due to its vulnerability against occlusions.

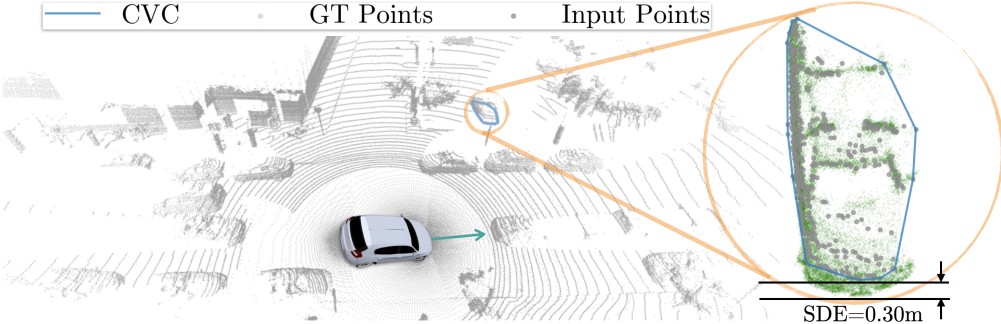

Figure 15: **Qualitative Results** showing the limitation of the convex visible contour (CVC). As depicted, due to the occlusion, CVC fails to cover the whole extent of the object. Note that we have visualized both the Lidar points from the current frame (in **gray**) as well as the aggregated points (in **green**) which are used to represent the true object shape.