# OpenReview forum: "Revisiting 3D Object Detection From an Egocentric Perspective"
_NeurIPS.cc/2021/Conference — NeurIPS 2021 Poster_

### Official Review · Reviewer_hKpm · 2021-07-16

**Rating:** 8
**Confidence:** 3

**Summary:**

This paper presents support distance error (SDE), a new evaluation metric 3D object detection from an egocentric point cloud. In contrast to Intersection over Union (IoU), SDE takes into account the potential collision induced by the object detector’s error over the planned trajectory of an ego-agent. It measures support distance (lateral and longitudinal directions) given the ego-agent’s orientation at given time. This metric is strongly indicative of collision risk.  Furthermore, the paper proposes StarPoly, a new object boundary representation that achieves better performance than traditional bounding box prediction on their new metrics SDE through experiments on autonomous driving Waymo Open dataset.

**Ethics Review Area:**

["I don’t know"]

**Limitations And Societal Impact:**

The paper mainly focuses on driving dataset, while the concepts can be applied to other types of datasets such as indoor object detection [1].

[1] K. Lai, L. Bo, X. Ren, and D. Fox, “A large-scale hierarchical multi-view rgb-d object dataset,” in Robotics and Automation (ICRA), 2011.


**Main Review:**

The ideas of SDE and StarPoly are simple yet more effective in collision measures than standard metric IoU on cuboids. This enables a proper selection of the object detection methods for downstream tasks, such as motion planning, collision avoidance, etc.

(+ Presentation) The paper is well-written in general. Schematic figures (1, 2, 5) are useful for understanding the concepts.

(+ Novel evaluation metric) The SDE-AP and SDE-APD metrics are derived from the longitudinal and lateral distances between the ego-agent and the object. Specifically, it depends on the boundary of the object and the planned trajectory of the ego-agent. The authors have shown in Section 3 that SDE has stronger correlation with the collision than the IoU.

(+ Novel boundary prediction) Due to the strong dependency on the object boundary of the SDE metric, the paper proposes to refine the standard cuboid shape bounding box using the StarPoly model. It takes as input a normalized point cloud extracted from the predicted bounding box and refines it such that it tightly covers the point cloud. With the StarPoly model, the paper shows a significant improvement in their new metric SDE (Section 5), allowing an ego-agent to better measure the potential collision.

(- Generalization) Unlike IoU, SDE is a metric scale metric. It is great for range measurements such as LiDar. However, in many computer vision applications, where physical camera calibration is not available, e.g., single view depth, this metric is not applicable. This is not a critical issue but it is worth highlighting or discussing the limitation of the metric and possible extension of SDE for such cases. And how does this can generalize to 3D SDE?


**Time Spent Reviewing:**

2

---

> ### Author Response · Authors · 2021-08-10
> **Responses to Reviewer hKpm**
>
> We sincerely appreciate the positive feedback from the reviewer hKpm and provide detailed responses below.
> ***
> **Q1: Unlike IoU, SDE is a scale metric. It’s not applicable when physical camera calibration is not available?**
>
> Thanks for the suggestion. We will include this discussion in the revision.
> ***
> **Q2: How does SDE generalize to 3D SDE?**
>
> We acknowledge that although 2D SDE is good enough for most driving cases, 3D SDE is needed for more challenging scenarios or other robot navigation tasks. There are a few straightforward ways to extend SDE to 3D. For example, one can evaluate the **minimal distance from all vertices of a 3D shape** to the longitudinal or lateral **plane** as the 3D SDE.
>
> ***
> **Limitations: The paper mainly focuses on driving datasets, while the concepts can be applied to other types of datasets such as indoor object detection [1].**
>
> We thank the reviewer for acknowledging the potential of our metrics and methods to be applied to other robotics tasks. We’ll add the reference to the dataset suggested and include discussion about this extension in the revision.

---

### Official Review · Reviewer_D4yU · 2021-07-17

**Rating:** 4
**Confidence:** 4

**Summary:**

The paper proposed 1) a new metric SDE measuring the error in predicted lateral and longitude distance from the ego-car to another car, 2) an object contour prediction model to refine the shape from a simple box. 3) SDE-based AP and weighted AP for evaluating object detectors.
The proposed SDE metric better reflects the risk of collision considering the ego-car trajectory.

**Ethical Concerns:**

No strong ethical concerns.

**Ethics Review Area:**

["I don’t know"]

**Limitations And Societal Impact:**

No strong limitations and negative social impacts.

**Main Review:**

Originality/Significance
1) The idea of SDE in lateral and longitude distances is relatively novel, making sense, and might be very practical for the purpose of collision prevention. However, as the authors mentioned that the sign of SDE has different meanings, I think we should keep the sign and use different thresholds for SDE-based AP. Note that it is riskier for positive SDE but less efficient for negative SDE.
Moreover, the examples in Fig. 4 (c,d) both have negative SDEs which means these tight boxes are only more conservative than a contour. Hence, I do not see a strong reason to compare different bounding box-based detectors using SDE-based AP. As shown in Fig. 7, the ranking of detectors is mostly unchanged (except [0m, 5m]). Moreover, we cannot see AP with respect to positive or negative SDE.
2) The proposed StarPoly is simple (limited novelty) and making sense considering SDE is computed using group truth aggregated point clouds. It is clear that StarPoly will increase SDE-based AP comparing to a box-based detector, but how critical is the improvement with respect to planning the ego agent to avoid a collision? I think an additional evaluation is needed to convince me.

To Summarized, SDE is an interesting new concept but without considering the sign it is limited. I am also not sure how SDE can be used to select a different box-based detector. Finally, do we really need to estimate the contour to avoid collision in most cases? Given the current evaluation, I am not convinced.

Quality/Clarity
The paper is easy to read with good illustrating figures.






**Time Spent Reviewing:**

3

---

> ### Author Response · Authors · 2021-08-10
> **Responses to Reviewer D4yU**
>
> We thank reviewer D4yU for the thoughtful comments. Please see our responses below.
> ***
> **Q1-a: I think we should keep the sign and use different thresholds for SDE-based AP.**
>
> That’s a good point. In this work, we intend to bring attention to the idea of egocentric evaluation. We propose SDE without sign as a simple implementation of this idea with minimal hyperparameters required. It is straightforward to extend it to more complicated versions with the sign included.
> We provide a plotting of [the distribution of signed SDE of detector boxes](https://ibb.co/pdPQnST) to understand the oversizing bias of box predictions.
> Based on specific requirements of an application, one can also have more fine-grained thresholds, e.g. different thresholds for positive and negative, and select the most suitable set up based on their priorities.
> ***
> **Q1-b: Tight boxes are only more conservative than contours in Fig. 4. Hence, no strong reason to compare box-based detectors using SDE-AP.**
>
> We’d like to reply to this concern in three-folds:
>
> * **[Equal importance of positive and negative SDE]** Although more conservative predictions seem safer intuitively, it can result in safety critical consequences in real-world autonomous driving scenes. For example, in narrow road scenarios, under-estimating the space available for driving can lead to undesirable and unsafe behaviors, such as hard braking (triggering rear end collisions), being stuck in the middle of the road, blocking the traffic and causing the trip to delay. Therefore, to achieve the highest safety standard, we aim to minimize both positive and negative SDE.
> * **[New capability of SDE-based AP]** Compared with IoU-based AP, SDE-based AP can more truthfully correlate the detection quality with collision predictions and therefore provides an essential support for evaluating the detection quality in planning, as shown in Tab. 1, Fig. 2, and Fig. 3.
> * **[New insights from SDE-based AP]** Using SDE-based metrics, we first reveal that boxes as detection output representation have inherent limitations (Fig. 4 c, d) and PointPillars outperforms MVF++ and PV-RCNN in the most safety-critical range, [0m, 5m]. These are not demonstrated in IoU-based metrics.
>
> ***
> **Q2: It is clear that StarPoly will increase SDE-based AP compared to a box-based detector, but how critical is the improvement with respect to planning the ego agent to avoid a collision?**
>
> Perception output is used in AV simulation to evaluate the behavior of a simulated ego-agent. We conducted further large-scale experiments in our internal dataset and found that using estimated amodal contours, compared to using detector boxes, leads to around **20% reduction in false positive (FP) collisions** while maintaining the same precision. This result indicates that our approach can significantly improve the evaluation of a planner system.
>
> More discussion regarding end-to-end AV stack evaluation is provided in our response to reviewer NNhf’s Q1-a and Q1-b (the second reviewer).

---

> > ### Comment · Reviewer_D4yU · 2021-09-04
> > **Response to author**
> >
> > Thanks for addressing my questions.
> > 1. Indeed sign and use different thresholds can be easily added. Please add the analysis in the revision.
> > 2. " 20% reduction in false positive (FP) collisions" actually means the bounding box is already pretty safe. Hence, I am still not sure how important it is to plan using StarPoly.
> > In general, the paper is very interesting and worth reading so I am fine with accepting this paper.

---

> > > ### Author Response · Authors · 2021-09-08
> > > **Response to Reviewer D4yU**
> > >
> > > Thanks for the suggestions. We will add the recommended sign and threshold analysis and add further discussion of when StarPoly is most valuable in comparison to bounding boxes.

---

### Official Review · Reviewer_NNhf · 2021-07-17

**Rating:** 6
**Confidence:** 4

**Summary:**

This paper presents a new egocentric measure for 3D object detection. The metric better captures the (local) object geometry and bettehr reflect the impact of detection quality on the ego-agent’s safety compared to IoU.

**Limitations And Societal Impact:**

The authors include the limitations of their model.

**Main Review:**

I'm currently on the fence (ie, borderline). I'd be happy to adjust my score provided my concerns are addressed.

---

**Strengths**
+ The paper is well-written and easy to follow
+ Literature review is comprehensive
+ Propose a new metric for 3D object detection with empirical analyses
+ Present a plug-n-play module that can be directly applied to existing detectors and boost their performance on the new metric

**Concerns**
- While the SDE metric may better reflect the possibility of collision, it is unclear whether the proposed metric/method is actually necessary/beneficial to the AV stack. In the main paper, the authors motivate SDE and hence the StarPoly merely through the collision of bbox. In practice, instead of directly treating the bbox as occupancy and plan a trajectory accordingly, state-of-the-art planners tend to exploit the perception results as guidance to estimate the cost map and then produce a trajectory. To be more specific, the bbox may be used to extract features, provide guidance on which part the model should attend to, or serve as regularization (through multi-task learning), but usually will not be used for explicit collision modeling. It would be thus more convincing if the proposed method is hooked up and evaluated with downstream planning/navigation tasks. Otherwise, even if a detector has better SDE-AP(D) (as shown in Sec. 5), we are still not sure if it really helps on navigation (which is the hope of such ego-centric metric).

- SDE, in some sense, captures the local geometry of the object (the part where it's used to compute the distance). Ideally the lower the SDE, the closer the shape is to the GT shape, the more consistent the collision estimation would be. To verify this, it would be great if the authors re-compute the numbers for Tab. 1, but this time using StarPoly to determine collision. Ideally the gap should be as obvious as using bbox.

- SDE is the max of latitude and longitudinal distance. Which one plays more important role? Intuitively I would guess latitude is more critical. Would be great if the authors could provide more analyses and insights.

- Why isn't  StarPoly in Tab. 2 GT column? Is StarPoly also inferior when using GT bbox?

- The output of StarPoly looks like a tighter rectangle with smooth corners and edges. is it really necessary to adopt a network to perform the task? how about just compute the mean shape/contour for all the vehicles, then during inference simply adjust the scale and directly apply? how does it work?

- The current CVC baseline is too weak. Seems a bit unfair to me. Vehicles has a very strong geometry prior - symmetry, and one should leverage it. For instance, use the longitudinal axis to flip (double) the point clouds and then compute CVC. This would provide a much better shape geometry. Furthermore, based on Fig. 10, the numbers of CVC looks pretty good, even better than StarPoly (3 out of 4 times). Is this a coincidence? If yes, would be great if the authors show the cases where CVC fails. Also, would be great if the authors could show which part of the geometry/contour is used to compute SDE in the images so that the reader can better understand what results in the error (something similar to Fig. 4).

- Why not simply use the points within the predicted bbox to represent the shape, estimate collision, and compute SDE metrics? Bbox are a coarse representation of the object geometry, while the surface points are more fine-grained. Can the authors comment about this?

- The authors should demonstrate the effectiveness with a real planner and showcase how the SDE based metrics transform to the downstream task comparing to IoU based metrics.

- L232: While the end results is the same, the explanation is a bit flawed. The barycenter loss is in fact encouraging x to lie on the same side as h wrt to the line formed by $v_r$ and $v_l$. It isn't forcing x to be in the triangle.

**Minor suggestions**
- I had a hard time understanding why we need SDE@t: since the authors adopt the GT rigid transform to warp the detection box, they are essentially assuming that the prediction error will remain the same (wrt the the GT box) throughout the time. It doesn't make much sense to me, why is this useful? Would be great if the authors could provide an example.

- The way StarPoly is parameterized restricted it to only handle non-convex shapes; the method is very similar to active contour methods (ie snake). The authors might want to cite a few papers there.

- I'm very confused how exactly is $SD_\alpha(B(o), e)$ computed at first. Based on Eq. 1 and the surrounding text (L137-L141), it is the minimum point-to-line distance among all the points *on the (GT) object boundary* wrt the lateral/longitudinal line. It is not clear what is the object boundary and what those points exactly are. It may be the object bbox and the points are sampled from the box boundary; or it may be the real object shape represented by LiDAR points enclosed by the bbox. In implementation details (L158-L161), the authors point out that aggregated LiDAR points are used to represent the (GT) shape. I thus assume its the latter for both. However, the illustrations in Fig. 1 and Fig. 4 suggest otherwise. Things are clearer when I read Sec. 4.2. Would be great if the authors make this more explicit. (I assume for CVC and StarPoly the points are also sampled from the boundary?)



**Time Spent Reviewing:**

7

---

> ### Author Response · Authors · 2021-08-10
> **Responses to Reviewer NNhf**
>
> We appreciate the detailed and high quality feedback from reviewer NNhf. Please see our detailed responses to the concerns (Q) and suggestions (S) below.
> ***
> **Q1-a: It is unclear whether the proposed metric is actually necessary/beneficial to the AV stack.**
>
> We think being able to accurately measure the collision risk is already very useful to AV development and evaluation. In AV simulation (replay of scenarios captured by AV's perception system), we need to measure the collision risks of different ego-agents’ behaviors to judge whether an AV algorithm is safe to deploy. Meanwhile, future trajectory predictions and motion planning commonly used by AV systems can benefit from collision risk estimations. The fidelity of such estimation largely depends on the perception quality. Using our proposed SDE-based metrics to measure perception quality can lead to more accurate understanding of the quality of the AV simulation results, compared to using non-egocentric metrics like IoU and IoU-AP.
>
> The proposed metrics lead us to develop better shape representations like the amodal contours. When the amodal contours are used in our internal collision detection model (for evaluating ego-agent’s behavior in simulation), it leads to 20% reduction in false positive collisions while maintaining the collision detection precision.
>
> ***
> **Q1-b: The authors should demonstrate the effectiveness with a real planner and showcase how the SDE based metrics transform to the downstream task compared to IoU based metrics.**
>
> While "end-to-end" evaluation metrics that connect perception to production-level driving performance are better conceptually, they would be very difficult to implement in practice.  Production AV systems have many components, including ones for motion planning, behavior forecasting, risk analysis, decision making, etc.  Those modules are not the same in different AV systems or even in different iterations of the same AV system, and any one of those components could introduce errors that mask differences in perception algorithms.  Comparing and tracking the performance of the perception component would be very challenging in this end-to-end setting -- comparisons only could be made if all the other components are both optimal and fixed.   So, instead, we aim to create a proxy metric that is independent of the other AV system components, but conceptually related to driving performance.
>
> This approach allows comparisons across systems and thus is common in the current AV research community and benchmark challenges (e.g. Waymo, Argo, nuScenes, etc.).  They all use proxy metrics (such as mIoU and IoU-AP on predicted bboxes) to measure perception quality independent of downstream components. In this work, we aim to provide improved proxy metrics that, while still decoupled from downstream components, are egocentric and more related to navigation tasks.
>
> ***
> **Q2: Re-compute the numbers in Tab. 1 using the StarPoly contours to determine collision.**
>
> Following the suggestion, we re-computed Tab. 1 using the contours drawn from our aggregated ground truth points, which should be the more accurate shape accessible. The results are as follows:
>
> | Measure | TP Mean | TP Median | FP/FN Mean | FP/FN Median |
> |---------|---------|-----------|------------|--------------|
> | IoU     | 0.902   | 0.911     | 0.904      | 0.903        |
> | SDE     | 0.114   | 0.095     | 0.161      | 0.153        |
>
> The gap between IoU and SDE is almost the same as the original Tab. 1 using boxes for collision tests.
> ***
> **Q3: SDE is the max of latitude and longitudinal distance. Which one plays a more important role?**
>
> In Tab. 1 of supplementary, we show the statistics of error contributions from $SDE_{lat}$ and $SDE_{lon}$ as follows (how often a large SDE is caused by lat or lon SD):
>
> |              | $SDE_{lat}$ | $SDE_{lon}$ |
> |--------------|--------------------|--------------------|
> | Contribution | 52%                | 48%                |
>
> It shows that $SDE_{lat}$ and $SDE_{lon}$ contribute almost equally in our LiDAR-based perception tasks (e.g. Waymo Open Dataset) while intuitively many would assume lateral weighs more than longitudinal.
> ***
> **Q4: Why isn't StarPoly in Tab. 2 GT column? Is StarPoly also inferior when using GT bbox?**
>
> No. StarPoly is better than GT bbox. We provide the detailed results here:
>
> |                    | Box   | CVC   | StarPoly |
> |--------------------|-------|-------|----------|
> | mSDE of [0m, 5m]   | 0.059 | 0.083 | 0.046    |
> | mSDE of [5m, 10m]  | 0.070 | 0.076 | 0.053    |
> | mSDE of [10m, 20m] | 0.094 | 0.142 | 0.068    |
> | mSDE of [20m, 40m] | 0.132 | 0.235 | 0.105    |
>
> We will include this result in the revision.
> ***
> **Q5:  Is it necessary to adopt a network to predict the contour? How do stronger baselines such as symmetric CVC and mean contour perform?**
>
> Although in the simple cases convex visible contour can do a good job to represent the boundary close to the ego agent, there are many other cases it may fail. For example, when the object is occluded (by another car or a pole), the visible contour is incomplete. Or when we care about the *amodal* quality of the contour, the visible contour is not able to recover the shape of the other side of the object.
>
> We thank the reviewer for proposing the two extra baselines: symmetric CVC (SymCVC) and mean contour (MeanContour). We provide an updated representation comparison as follows:
>
> | Representation | 5F-MVF++ | MVF++ | PV-RCNN | PointPillars |
> |----------------|----------|-------|---------|--------------|
> | Box            | 0.874    | 0.834 | 0.807   | 0.817        |
> | CVC            | 0.874    | 0.865 | 0.865   | 0.847        |
> | SymCVC         | 0.889    | 0.878 | 0.879   | 0.858        |
> | MeanContour    | 0.892    | 0.878 | 0.865   | 0.860        |
> | StarPoly       | 0.924    | 0.913 | 0.896   | 0.892        |
>
> We find that StarPoly still consistently outperforms all the other 4 baseline representations. We will include these new baselines in the revision.
> ***
> **Q6: Explain why CVC looks good in Fig. 10 but not in tables? Show failure cases like Fig. 4.**
>
> CVC is a heuristic method fully depending on **the visible point cloud** of each object. It therefore has two major drawbacks (as explained in the reply to the previous question)
> * It is very sensitive to occlusions, which are common in Autonomous Driving.
> * It is mostly partial. Hence, it’s not suitable for future planning where we may come close to currently occluded parts. A sufficiently complete amodal shape estimation is needed to handle such cases.
>
> StarPoly can address such issues due to its amodal nature and data-driven learning. In Fig. 10, as we only zoomed into the closest 4 vehicles without occlusions, the CVC performs well.
>
> In Fig.4 of supplementary, we showcase one representative failure of CVC due to occlusion. The SDE of CVC is as high as 0.3m. Moreover, in Fig. 3 of supplementary, we demonstrate the limitation of CVC in future planning.
> ***
> **Q7: Why not simply use the points within the predicted bbox to represent the shape, estimate collision, and compute SDE metrics?**
>
> The CVC baseline, being the convex hull of the points within the predicted bbox, is following this idea although being enhanced by enforcing convexity. One can see CVC’s performance as the proxy of directly using the points in the bounding box.
> ***
> **Q9: L232: While the end results are the same, the explanation is a bit flawed.**
>
> You are right. $\mathcal{L}_{c}$ is encouraging $x$ to lie on the interior side of line $v_r v_l$. We’ll clarify this in the revision.
> ***
> **S1: Provide an example on why we need SDE@t.**
>
> Please see our general motivation of SDE@t in our response above to Reviewer irMp’s (the first reviewer) question N3.
>
> We provide one demonstrative example in Fig. 3 of the supplementary. In this case, the ego-agent started from driving behind an object and ended up overtaking it. To plan for this overtaking path, it’s important to know what the front of the object may look like even when only its back is visible to the ego-agent at t=0. SDE@t is hence suitable for this assessment. For instance, it reveals the limitation of a very partial CVC (low SDE@0 but high SDE@8).
> ***
> **S2: StarPoly is restricted to only non-convex shapes; It’s similar to active contour methods (i.e. snake), which should be cited.**
>
> By design StarPoly is able to reconstruct both convex and non-convex shapes. For instance, a square (convex) and a star shape (non-convex) can both be represented by StarPoly. We agree that snakes are an alternative approach to contour detection. We will add references to active contour methods in our revision.
> ***
> **S3: It is not clear what $SD_{\alpha}(B(o), e)$ is computing.**
>
> By definition, $B(o)$ is the point set on the boundary of object $o$. Therefore, this defines the minimal distance from the boundary to lines (lateral or longitudinal). Note that this general definition applies to both objects represented directly by their boundaries (CVC and StarPoly) and objects represented by point sets w/ or w/o interior points included (ground truth points).
> In implementation, we compute $SD$ of ground truth points by taking the minimal point-to-line distance of all points, including interior points. This is a simplified implementation without the need of extracting boundaries. We will clarify this in the revision.

---

### Official Review · Reviewer_irMp · 2021-07-28

**Rating:** 6
**Confidence:** 4

**Summary:**

This paper proposes a novel ego-centric evaluation criteria for 3D object detection. The proposed metric, SDE, measures error in 3D object detection prediction in terms of lateral and longitudinal distances w.r.t. the ego vehicle. Moreover, the paper proposes a novel approach to represent objects as predict amodal contours, which is used to improve the proposed SDE metric on the Waymo Open dataset.

**Ethical Concerns:**

No.

**Limitations And Societal Impact:**

This has been talked about to some extent.

**Main Review:**

**Positives**
P1. The paper proposes a metric which incorporates two properties: (a) evaluating 3D object detection prediction w.r.t. the ego vehicle, which is directly correlated to the chances of collision w.r.t. the ego vehicle. (b) defining lateral and longitudinal distance supports to capture the parts of the boundary which are closer to the ego-agent.
P2. The paper proposes a new optimization approach which predicts more tightly fitted amodal contour outlines of the detected LIDAR bounding boxes.
P3. The paper analyzes the usefulness of the metric and the StarPoly approach with their metric on the Waymo Open dataset.

**Negatives**
N1. While the proposed SDE metric is novel in two aspects (P1a and P1b), it would be great if the paper analyzes the metrics by breaking them down one by one. This can be shown in Table 1 and Fig. 2, while comparing the proposed SDE metric w.r.t. IoU. While IoU is not an ego-centric measure, it would be interesting to define an ego-centric version of IoU measure which can help us determine how well a 3D bounding box prediction w.r.t. ego vehicle correlates with CDA? This would break down the importance of defining an ego-centric measure as well as focusing on the parts of the boundary which are more important from the perspective of the ego vehicle.
N2. Should there be a normalization factor for the definition of $ SD_{\alpha} $ (Eq. 2), especially when $ \alpha = lon $? Larger distances to these support distance measures mean that the difference w.r.t. GT would be higher (Eq. 2). This would impact definition of SDE, as the minimum of its lateral and longitudinal measures. Hence it is not clear whether those two support distance measures are on the same scale or not.
N3. It is not clear what is the purpose of proposing SDE@t metric? Besides a couple of plots to motivate certain aspects of the paper, it is not clear what is the usefulness of this measure and the understanding we gain from this?


**Questions**
Some things weren't clear.
Q1. In Fig. 3, what is CDA? What does mean correspond to here? Is it mean across all the object instances?
Q2. In l.164-165, it is mentioned that SDE_{lat} is very useful indicator to reflect collision risk. But isn't SDE_{lon} also a useful indicator, especially for head-on collisions?
Q3. In l.137, while defining "longitudinal line", shouldn't the line be passing through the center as well?
Q4. Why is GT box column (col 5) in Table 2 not 0? Doesn't Eq 2 collapse to 0 here?
Q5. While comparing with [33] in supplementary material, it is mentioned that one of the benefits of this approach is stability. It is not clear how this is an advantage. Since the proposed ego-centric measure focuses on correlation with the other objects, such 3D bounding box detections will always be tied with a planner. While [33] uses a particular predicted planner, this approach relies on the ground truth rigid body motion. So essentially this approach also relies on a particular planner algorithm. It is not clear why this approach which propagates errors in its planner algorithm better than [33]. In terms of stability, while measuring one detection algorithm to another, the planner component can always be kept constant. This point is not quite clear.
Q6. In the analysis of the SDE metric, it would be interesting to see how the human annotation of the bounding boxes "agree" with the SDE metric?
Q7. Does the StarPoly approach gives better bounding box predictions in terms of the IoU metric?
Q8. It is mentioned in l.208 that the bounding box is expanded for input to the StarPoly PointNet model. It would be good to have an ablation study which can show how this factors into the proposed algorithm. If the StarPoly algorithm is not very sensitive to this, wouldn't different algorithms with different detector behavior can have similar outcomes with the StarPoly algorithm? If the StarPoly algorithm is indeed sensitive to this, it would be interesting if the authors talked about how this algorithm is sensitive to the expansion and also the different detection algorithms. While, the latter is shown to some extent in the paper, but it would be useful to analyze (quantitatively and qualitatively) the corresponding StarPoly predictions from the different behaviors of the different detector algorithm,

**Time Spent Reviewing:**

5-6

---

> ### Author Response · Authors · 2021-08-10
> **Responses to Reviewer irMp**
>
> We appreciate the detailed and high quality feedback from reviewer irMp. Please see our detailed responses below.
>
> ***
> **N1: It would be great if the paper analyzes the metrics by breaking down the properties P1a and P1b one by one. While IoU is not an ego-centric measure, it would be interesting to define an ego-centric version of IoU measure which can help us determine how well a 3D bounding box prediction w.r.t. ego vehicle correlates with CDA?**
>
> We interpret there could be two directions to address the comments and prepared two versions of replies. We would appreciate further clarification/feedback from the reviewer based on our response.
>
> Case 1: The comment is related to the distance weighting in computing APD (distance weighted Average Precision).
>
> In Tab. 1 of the paper, we compare SDE with IoU, where SDE shows a stronger correlation with collision detection accuracy. Note that the statistics in Tab. 1 don’t include any distance weighting, hence they directly reflect the importance of SDE.
> In our paper, we introduce distance weighting as a simple proxy of distance breakdowns as in Fig. 7. Based on your question, we extend IoU-AP to IoU-APD with the same distance weighting as SDE-APD, to study the importance of distance weighting. The results are as follows. Note that while IoU-APD and IoU-AP have the same ordering, SDE-APD is able to reveal a different ranking between PointPillars and PV-RCNN.
>
> | Method       | IoU-AP | IoU-APD | SDE-APD |
> |--------------|--------|---------|---------|
> | 5F-MVF++     | 0.863  | 0.989   | 0.874   |
> | MVF++        | 0.814  | 0.981   | 0.834   |
> | PV-RCNN      | 0.797  | 0.972   | 0.808   |
> | PointPillars | 0.720  | 0.966   | 0.817   |
>
> Case 2: The comment is related to the shape measure (irrelevant to false positive and false negative nor distance weighting in computing the Average Precision).
>
> In this case, an ego-centric IoU measure means we weight the IoU computation based on how close a box sub-area is to the visible boundary. However, it is not clear how to formally define such a measure with clear physical meaning (as in the lateral and longitudinal distances).
>
> While we have demonstrated that SDE (using boxes as contours) has a stronger correlation with collision rate compared to the box IoU in Tab. 1, we leave comparisons to more alternative designs (e.g. boundary weighted IoU) to future work.
>
> ***
> **N2: Should there be a normalization factor to balance lateral and longitudinal SD scales (esp. lon)?**
>
> In Tab. 1 of supplementary, we show the statistics of $SDE_{lat}$ and $SDE_{lon}$:
>
> | Statistics | $SDE_{lat}$ | $SDE_{lon}$ |
> |------------|--------------------|--------------------|
> | Mean (m)   | 0.17               | 0.17               |
> | Median (m) | 0.12               | 0.11               |
>
> We find that $SDE_{lat}$ and $SDE_{lon}$ are at the same scale so we don’t use normalization for our experiments.
> For practical use cases, a user can extend the metrics with different thresholds for the two terms to address their preferences of error tolerance.
>
> ***
> **N3: It is not clear what the purpose of SDE@t is.**
>
> While SDE (in default at t=0s) measures the distance errors of the detector contours/boxes to the **current** ego agent position, SDE@t measures the distance errors to a **future** position of the ego agent (using observed future trajectories for the objects).
>
> SDE@t is useful in cases such as cut-in and turning, where the ego agent will travel to the other side or get close to the back of the objects. In those cases, even if the SDE@t=0 is accurate, the SDE@t may have larger errors if the unobserved boundary is not accurately estimated. Getting a small SDE@t is therefore important for the ego agent to plan a safe path for those cases.
>
> Fig. 5 in the main paper shows that using SDE@t we can compare how different shape representations (boxes, visible contours, amodal contours from StarPoly) may impact **future** trajectory planning (distance errors to the future ego-agent positions).
> Fig. 3 in the supplementary gives an illustration of a case of large SDE@t error due to poor contour estimation from the visible contours.
>
> ***
> **Q1: What is CDA in Fig. 3?**
>
> CDA in Fig.3 is the collision detection accuracy (at current and future times). It is computed as the portion of collision detections using predicted boxes (by transforming the current frame prediction box to the future frame) versus using ground truth boxes (at the future frame). Naturally, as the current frame prediction has limited accuracy in recovering the unseen part, the CDA becomes worse as we evaluate it into the future.
> The “mean” in mIoU and mSDE stands for the average over all object instances. We’ll add clarifications in the revision.
>
> ***
> **Q2: L164-165: isn’t $SDE_{lon}$ also a useful indicator?**
>
> Yes it is. In our evaluation metrics, we compute both $SDE_{lat}$ and $SDE_{lon}$. Although we only mentioned $SDE_{lat}$ in the text, we illustrated $SDE_{lon}$ in Fig. 1 as well. Moreover, we quantitatively studied the contribution of these 2 terms in Tab. 1 of supplementary and showed their equal importance.
>
> ***
> **Q3: L137: shouldn’t the “longitudinal line” be passing through the center as well?**
>
> Yes. We’ll clarify this in the revision.
>
> ***
> **Q4: Why GT box in Table 2 not 0? Doesn’t Eq 2 collapse to 0 here?**
>
> Please note that we use **aggregated GT points** to represent the true object boundary (i.e. $B_{gt}$) as described in L158-L159. Thus, using ground truth **Box** or **CVC** won’t yield a zero SDE.
>
> ***
> **Q5: It’s not clear how the stability of SDE is an advantage?**
>
> We thank the reviewer for this thoughtful comment. We address it in two aspects.
>
> First, when we consider SDE itself, as a shape metric, it is not dependent on any planner. It can be measured in a *static* scene given a known sensor position (i.e. the ego agent position). This however is not achievable with the [33] which will always rely on a planner (and usually it has to be a ML-based one).
>
> Second, when we consider a dynamic scene and would like to evaluate the SDE@t (see reply to N3 for the motivation of it), we would like a "stable" set of ego-agent and object trajectories to evaluate our metric. We could pick a particular planner algorithm, like [33], and evaluate detections only for that planner. However, in a production setting, when detectors and planners are being developed concurrently using a mixture of ML models, search, and optimization, it would be difficult to compare detection results across different iterations of the system. Nevermind that a specific planner may require different input representations (contours, boxes, occupancy maps). So, we propose to use observed trajectories to compute SDE@t, which do not change (are stable) as the system is being developed. We will add more clarifications to the revised supplementary.
>
> ***
> **Q6: How does the human annotation of the bounding boxes “agree” with the SDE metric?**
>
> In Tab. 2 of the paper, the Ground Truth/Box column lists the SDE of human annotated **boxes** using ground truth **points**. It shows how well these **GT Boxes** agree with the actual object shape in terms of SDE.
>
> ***
> **Q7: Does StarPoly give better bbox in terms of IoU?**
>
> In a new experiment to answer this question, we draw tight bounding boxes of StarPoly’s contour predictions and compute their IoU-AP and IoU-APD:
>
> | Method       | IoU-AP | IoU-APD |
> |--------------|--------|---------|
> | Detector Box | 0.797  | 0.972   |
> | StarPoly Box | 0.772  | 0.968   |
>
> We see that StarPoly’s boxes are slightly worse than detector boxes on average, which is expected as we don’t optimize contours to get better boxes. Moreover, we have seen in some challenging cases (e.g. a construction vehicle with an extruding arm or a SUV with some trailer) that StarPoly can help to correct the detection box. One of the future steps is to further evaluate on those challenging cases and improve the optimization of StarPoly to enhance box quality in general.
>
> ***
> **Q8: Ablation on different box expansions on the bbox for StarPoly.**
>
> In a new experiment to answer this question, we provide an ablation on using different levels of box expansion with the PV-RCNN detector:
>
> | Expansion (m) | 0.0   | 0.3   | 0.6   | 1.2   |
> |---------------|-------|-------|-------|-------|
> | SDE-APD       | 0.889 | 0.896 | 0.887 | 0.871 |
>
> We find that StarPoly is not sensitive to different expansion rates.
> We have also shown in Fig. 8 of the main paper that StarPoly consistently improves the egocentric performances based on different detectors. However, due to different precision/recall rate of those detectors, StarPoly would not yield the same SDE-APD.

---

### Author Response · Authors · 2021-08-10
**General Responses**

We thank all reviewers for their time and effort spent providing the valuable comments and constructive feedback.

We are glad to see that there are some shared positive feedback on our paper:
* A novel ego-centric evaluation metric for 3D object detection (all).
* A novel plug-n-play approach, StarPoly, to predict tightly fitted amodal contours and boost the performance of existing detectors (R irMp, R NNhf, and R hKpm).
* Paper is well-written and easy to follow (all).

We believe this work will help draw more attention from the Perception community to rethink the goals and evaluation protocols for common computer vision tasks (3D detection for this work), esp. when perception is used to serve downstream applications like robotics and autonomous driving.

Please see the individual threads for responses to each reviewer.

---

### Decision · Program_Chairs · 2021-09-27

**Decision:**

Accept (Poster)

**Comment:**

After the rebuttal most reviewers agreed to accept this paper. The remaining concerns centered around connections of the proposed evaluation method and algorithm to down-stream planning and driving performance. The AC agrees that the paper would be stronger making these connections, but sees enough merit in the proposed work to accept the paper without it.